# Redundant representations help generalization in wide neural networks

**Diego Doimo**[*]
International School for Advanced Studies

**Aldo Glielmo**
International School for Advanced Studies
Bank of Italy[†]

**Sebastian Goldt**
International School for Advanced Studies

**Alessandro Laio**
International School for Advanced Studies

## Abstract

Deep neural networks (DNNs) defy the classical bias-variance trade-off: adding parameters to a DNN that interpolates its training data will typically improve its generalization performance. Explaining the mechanism behind this "benign overfitting" in deep networks remains an outstanding challenge. Here, we study the last hidden layer representations of various state-of-the-art convolutional neural networks and find that if the last hidden representation is wide enough, its neurons tend to split into groups that carry identical information and differ from each other only by statistically independent noise. The number of such groups increases linearly with the width of the layer, but only if the width is above a critical value. We show that redundant neurons appear only when the training is regularized and the training error is zero.

## 1  Introduction

Deep neural networks (DNN) have enough parameters to achieve zero training error, even with random labels [1, 2]. In defiance of the classical bias-variance trade-off, the performance of these *interpolating classifiers* improves as the number of parameters increases well beyond the number of training samples [3–6]. Despite recent progress in describing the implicit bias of stochastic gradient descent towards "good" minima [7–12], and the detailed analysis of solvable models of learning [13–21], the mechanisms underlying this "benign overfitting" [22] in deep neural networks remain unclear, especially since their loss landscape contains "bad" local minima and SGD can reach them [23].

In this paper, we describe a phenomenon in wide DNNs that could be a possible mechanism for benign overfitting when the networks are trained with regularization. We illustrate this mechanism in Fig. 1 for a family of increasingly wide DenseNet40s [24] trained on CIFAR10 [25] following common practice, in particular using weight decay (see Sec. 2.1). For simplicity, we refer to the width $W$ of the last hidden representation as the width of the network. The blue line in Fig. 1-b shows that the average classification error (error) approaches the performance of a large ensemble of networks ($\text{error}_\infty$) [21] as we increase the network width $W$. In agreement with [26], we find that the performance of these DenseNets improves continuously with width. For widths greater than 350, the networks are wide enough to reach zero training error (see Appendix, Sec. B, Fig S2-c) and,

---

[*]ddoimo@sissa.it

[†]The views and opinions expressed in this paper are those of the authors and do not necessarily reflect the official policy or position of Bank of Italy.

36th Conference on Neural Information Processing Systems (NeurIPS 2022).

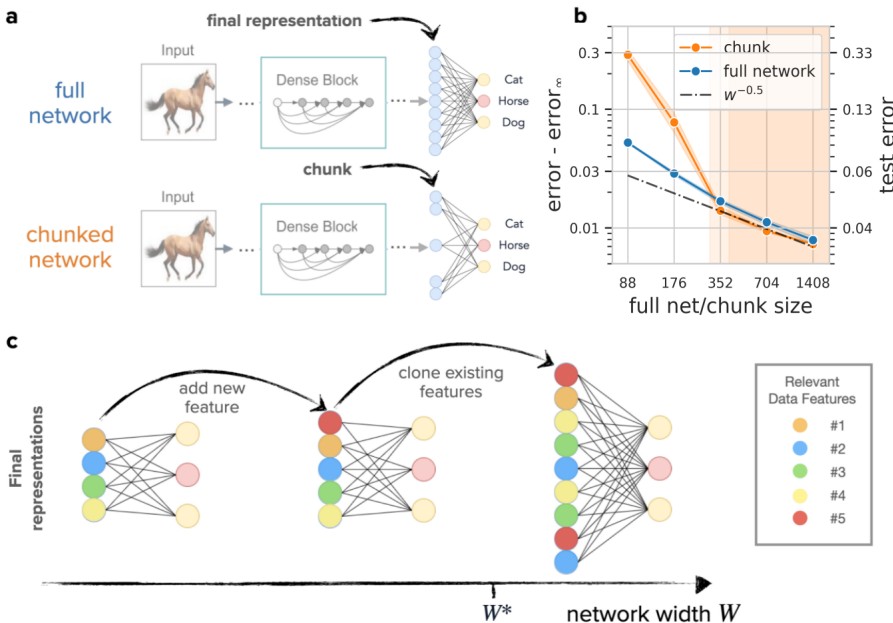

Figure 1: **The redundancy of representations in wide neural networks. a**: We analyze the final representations of deep neural networks (DNN), namely the activities of the last hidden layer of neurons (light blue) We focus on the performance and the statistical properties of randomly chosen subsets of $w_c$ neurons which we call "*chunks*". In the chunked network shown here, $w_c = 5$ out of 9 neurons are kept and used to predict the output. **b**: As we increase the size of the chunk $w_c$ that we keep in a state-of-the-art DNN, here a DenseNet40, the test error of the chunk (orange line) becomes similar to the test error of a full network of width $W = w_c$ (blue line). In this regime, which is reached when $w_c$ is larger than a threshold $w_c^*$ (shaded area) the error approaches its asymptotic value $\text{error}_\infty$ as a power-law $w_c^{-1/2}$ (dashed line). $\text{error}_\infty$ is the error of an ensemble average of 20 networks of the widest size. **c**: Illustration of three final representations for networks of increasing width. In small networks, an additional neuron fits new features of the data (red neuron). As the network width goes beyond a critical width $W^*$, additional neurons instead copy features already learned from data, instead of over-fitting to features that are not relevant to the task. This mechanism is suggested by the $w_c^{-1/2}$ decay of the chunk error, and by the statistical analysis, we present in this paper.

interestingly, their test error decays approximately as $W^{-1/2}$. Our goal is to understand how the error of the network can keep decaying beyond the interpolation threshold, and why it decays as $W^{-1/2}$.

We make our key observation by performing the following experiment: we randomly select a number $w_c$ of neurons from the last hidden layer of the widest DenseNet40 and remove all the other neurons from that layer as well as their connections (Fig. 1-a). We then evaluate the performance of this chunk of $w_c$ neurons, *without* retraining the network. The orange profile of Fig. 1-b shows the test error of *chunks* of varying sizes. There are two regimes: for small chunks, the error decays faster than $w_c^{-1/2}$, while beyond a critical chunk size $w_c^*$ (shaded area), the error of a chunk of $w_c$ neurons is roughly the same as the one of a full network with $w_c$ neurons. Furthermore, the error of the chunks decays with the same power-law $w_c^{-1/2}$ beyond this critical chunk size.

The decay rate of $-1/2$ suggests that in this regime chunks of $w_c$ neurons can be thought of as statistically independent estimators of the same features of the data, differing only by small, uncorrelated noise. In other words, beyond the critical width $w_c^*$, the final hidden representation of an input in a trained, wide DNN becomes highly redundant. This motivates a possible mechanism for benign overfitting, schematically portrayed in Fig. 1-c: as the network becomes wider, additional neurons are first used to learn new features of the data. Beyond the critical width $w_c^*$, additional neurons in the final layer don't fit new features in the data, and hence over-fit; instead, they make a copy, or a *clone*, of a feature that is already part of the final representation. The last layer thus splits into more

and more clones as the network grows wider as we illustrate at the bottom of Fig. 1. The accuracy of these wide networks then improves with their width because the network implicitly averages over an increasing number of clones in its representations to make its prediction.

This paper provides a quantitative analysis of this phenomenon on various data sets and architectures. Our main findings can be summarized as follows:

1. A chunk of $w_c$ random neurons of the last hidden representation of a wide neural network predicts the output with an error that decays as $w_c^{-1/2}$ if the layer is wide enough and $w_c$ is large enough. In this regime, we call the chunk a "clone";

2. Clones fit the training set with zero error and can be linearly mapped one to another, or to the full representation, with an error that can be described as uncorrelated random noise.

3. Clones appear if the model is trained with weight decay and the training set is fitted with zero error. If training is stopped too early or if the training is performed without regularization, 1. and 2. do not take place, even if the last representation is very wide.

## 2 Methods

### 2.1 Neural network architectures

We report experimental results obtained with several architectures (fully connected networks, Wide-ResNet-28, DenseNet40, ResNet50) and data sets (CIFAR10/100 [25], ImageNet [27]). We train all the networks using SGD with momentum and, importantly, weight decay. The amount of weight decay is found with a small grid search, while the other relevant hyperparameters are set following standard practice. We give detailed information on our training setups in Sec. A of the Appendix. All our experiments are run on Volta V100 GPUs. In the following paragraphs, we describe how we vary the width $W$ of the models.

**Fully-connected networks on MNIST.** We train a fully-connected network to classify the parity of the MNIST digits [28] (pMNIST) following the protocol of Geiger *et al.* [21]. MNIST digits are projected on the first ten principal components, which are then used as inputs of a five-layer fully-connected network (FC5). The four hidden representations have the same width $W$ and the output is a real number whose sign is the predictor of the parity of the input digit.

**Wide-ResNet-28 and DenseNet40 on CIFAR10/100.** We train CIFAR10 and CIFAR100 on family of Wide-ResNet-28 [26] (WR28). The number $W$ of the last hidden neurons in a WR28-$n$ is $64 \cdot n$, obtained after average pooling the last $64 \cdot n$ channels of the network. In our experiments, we also analyze two narrow versions of the standard WR28-1 which are not typically used in the literature. We name them WR28-0.25 and WR28-0.5 since they have 1/4 and 1/2 of the number of channels of WR28-1. Our implementation of DenseNet40 follows the DenseNet40-BC variant [24]. We vary the number of input channels $c$ in $\{16, 32, 64, 128, 256\}$, which is twice the growth rates $k$ of the networks [24]. The number $W$ of the last hidden features of this architecture is $5.5 \cdot c$.

**ResNet50 on ImageNet.** We modify the ResNet50 architecture [29] by multiplying by a constant factor $c \in \{0.25, 0.5, 1, 2, 4\}$ the number of channels of all the layers after the input stem. When $c = 2$ our networks differ from the standard Wide-ResNet50-2 [26] since we double the channels of *all* the layers and not just those of the bottleneck of the ResNet blocks. As a consequence in our implementation, the number of features after the last pooling layer is $W = 2048 \cdot c$ while in [26] $W$ is fixed to 2048.

### 2.2 Analytical methods

**Reconstructing the wide representation from a smaller chunk.** To determine how well a subset of $w$ neurons can reconstruct the full representation of size $W$ we search for the $W \times w$ linear map $\mathbf{A}$, able to minimize the squared difference $\left(\mathbf{x}^{(W)} - \hat{\mathbf{x}}^{(W)}\right)^2$ between the $W$ activations of the full layer representation, $\mathbf{x}^{(W)}$, and the activations predicted from the chunk of size $w$, $\hat{\mathbf{x}}^{(W)}$:

$$\hat{\mathbf{x}}^{(W)} = \mathbf{A}\mathbf{x}^{(w)}. \tag{1}$$

This least-squares problem is solved with ridge regression [30] with regularization set to $10^{-8}$, and we use the $R^2$ coefficient of the fit to measure the predictive power of a given chunk size. The $R^2$ value is computed as an average of the single-activations $R^2$ values corresponding to the $W$ output coordinates of the fit, weighted by the variance of each coordinate. We further compute the $W \times W$ covariance matrix $C_{ij}$ of the *residuals* of this fit, and from $C_{ij}$ we obtain the correlation matrix as:

$$\rho_{ij} = \frac{C_{ij}}{\sqrt{C_{ii}C_{jj} + 10^{-8}}}, \tag{2}$$

with a small regularization in the denominator to avoid instabilities when the standard deviation of the residuals falls below machine precision. To quantify how much the errors of the fit are correlated, we average the absolute values of the non-diagonal entries of the correlation matrix $\rho_{ij}$. For short, we refer to this quantity as a 'mean correlation'.

**Reproducibility.** We provide code to reproduce our experiments and our analysis online at `https://github.com/diegodoimo/redundant_representation`.

## 3    Results

**The test error of chunks of $w_c$ neurons of the final representation asymptotically scales as $w_c^{-1/2}$.** The mechanism we propose is inspired by the following experiment: we compute the test accuracy of models obtained by selecting a random subset of $w_c$ neurons from the *final hidden representation* of a wide neural network. We select $w_c$ neurons at random and we compute the test accuracy of a network in which we set to zero the activation of all the other $w - w_c$ neurons of the final layer. Importantly, we do not fine-tune the weights after selecting the $w_c$ neurons: all the remaining parameters of the previous layers are left unchanged and only the the activations of the "killed" neurons of the last hidden representation are not used to compute the logits. We take 500 random samples of neurons for each chunk width $w_c$. We consider three different data sets: pMNIST trained on a fully connected network, CIFAR10 and CIFAR100 trained on convolutional networks. The width $W$ of the network is 512 for pMNIST and CIFAR10, and $W = 1024$ for CIFAR100 (see Sec. 2.1). In all these cases, $W$ is large enough to be firmly in the regime where the accuracy of the networks scales (approximately) as $W^{-1/2}$ (see Fig. 2).

In Fig. 3 we plot the test error of the "chunked models" as a function of $w_c$ (orange lines). The behavior is similar in

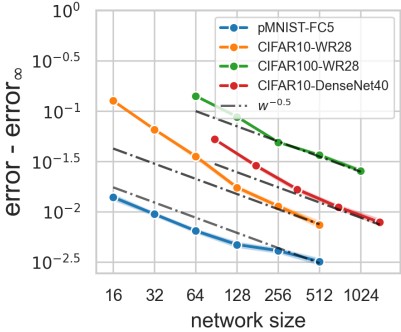

Figure 2: **Scaling of the test error with width for various DNN.** The average test error of neural networks with various architectures approaches the test error of an ensemble of such networks as the network width increases. The network size shown here is the width of the final representation. For large width, we find a power-law behavior error $-$ error$_\infty \propto W^{-1/2}$ across data sets and architectures. Full experimental details in Sec. 2.1

all three networks: the test error decays as $w_c^{-1/2}$ for chunks that are larger than a critical value $w_c^*$, which depends on the data set and architecture used. This decay follows the same law observed for full networks of the same width (Fig. 2). This implies that a model obtained by selecting a random chunk of $w_c > w_c^*$ neurons from a wide final representation behaves similarly to a full network of width $W = w_c$. Furthermore, a decay with rate $-1/2$ suggests that the final representation of the wide networks can be thought of as a collection of statistically independent estimates of a finite set of data features relevant for classification. Adding neurons to the chunk hence reduces their prediction error in the same way an additional measurement reduces the measurement uncertainty, leading to the $-1/2$ decay.

At $w_c$ smaller than $w_c^*$ instead, the test error of the chunked models decays faster than $w_c^{-1/2}$ in all the cases we considered, including the DenseNet architecture trained on CIFAR10 shown in Fig. 1-b. In this regime, adding neurons to the final representation improves the quality of the model significantly quicker than it would in independently trained models of the same width (see Fig. 1-c for a pictorial representation of this process). We call chunks of neurons of size $w_c \geq w_c^*$ *clones*. In the following, we characterize more precisely the properties of the clones.

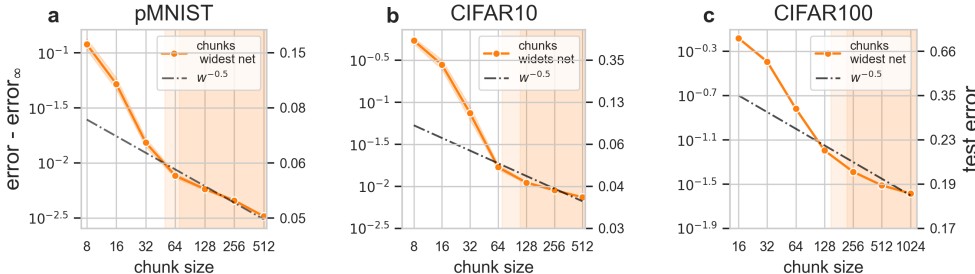

Figure 3: **Scaling of the test error of chunks of neurons extracted from the final hidden representation of wide NNs.** We plot how the test error of chunked networks approaches $\text{error}_\infty$, the error of an ensemble of 20 networks of the widest size (e.g. $W = 1024$ for CIFAR100), as the chunk size $w_c$ increases. Chunks are formed by selecting a number of $w_c$ neurons at random from the final hidden representation of the widest networks: a FC5 on pMNIST (width $W = 512$), and Wide-ResNet-28 for CIFAR10 ($W = 512$) and CIFAR100 ($W = 1024$). The shaded regions indicate regions where the error of the chunks with $w_c$ neurons decays as $w_c^{-1/2}$.

**Clones interpolate the training data.** A trained deep network often represents the salient features of the data set well enough to achieve (close to) zero classification error on the training data. In the top panels of Fig. 4, we show that wide networks can interpolate their training set also using just a subset of $w_c > w_c^*$ random neurons: the dark orange profiles show that when the size of a chunk is greater than $w_c^* \sim 50$ for pMNIST, 100 for CIFAR10 and 200 for CIFAR100, the predictive accuracy on the training set remains almost 100%. The minimal size of a clone $w_c^*$ can be identified with the minimal number of neurons required to interpolate the training set. Beyond $w_c^*$, the neurons of the final representation become redundant since the training error remains (close to) zero even after removing neurons from it. The number of distinct clones in a network of width $W$ is $n = W/w_c^*$. If distinct clones provide independent measures of the same salient features of the data, the *test* error decays approximately as $n^{-1/2}$ or equivalently $W^{-1/2}$. In the following, we will indeed see that distinct clones differ from each other by uncorrelated random noise.

**Clones reconstruct the full representation almost perfectly.** From a geometrical perspective, the important features of the final representation correspond to directions in which the data landscape shows large variations [31]. A clone is a chunk that is wide enough to encode almost exactly these directions (since its training error is almost zero), but using much fewer neurons than the full final representation. We analyze this aspect by performing a linear reconstruction of the $W$ activations of the last hidden representation of the widest network starting from a random subset of $w_c$ activations using ridge regression with a small regularization penalty according to Eq. (1). The blue profiles in Fig. 4-(d,e,f), show the $R^2$ coefficient of fit as a function of the chunk size $w_c$ for pMNIST (left), CIFAR10 (center), CIFAR100 (right). When $w_c$ is very small, say below 6 for pMNIST, 20 for CIFAR10 and 60 for CIFAR100, the $R^2$ coefficient grows almost linearly with $w_c$ [3]. In this regime, adding a randomly chosen activation from the full representation to the chunk increases substantially $R^2$. When $w_c$ becomes larger $R^2$ reaches almost one. This transition happens when $w_c$ is still much smaller than $W$ and corresponds approximately to the regime in which the test error starts scaling with the inverse square root of $w_c$ (see Fig. 3). The almost perfect reconstruction of the original data landscape with few neurons is a consequence of the low *intrinsic dimension* (ID) of the representation [32]. The ID of the widest representations gives a lower bound on the number of coordinates required to describe the data manifold, and hence on the neurons that a chunk needs in order to have the same classification accuracy as the whole representation. The ID of the last hidden representation is 2 in pMNIST, 12 in CIFAR10, 14 in CIFAR100, numbers which are much lower than $w_c^*$, the width at which a chunk can be considered a clone.

**Clones differ from each other by uncorrelated random noise.** When $w_c > w_c^*$ the small residual difference between the chunked representation and the full representation can be approximately

---

[3]The linear trend can not be clearly seen in Fig. 4 as we plot the $x$-axis with a logarithmic scale.

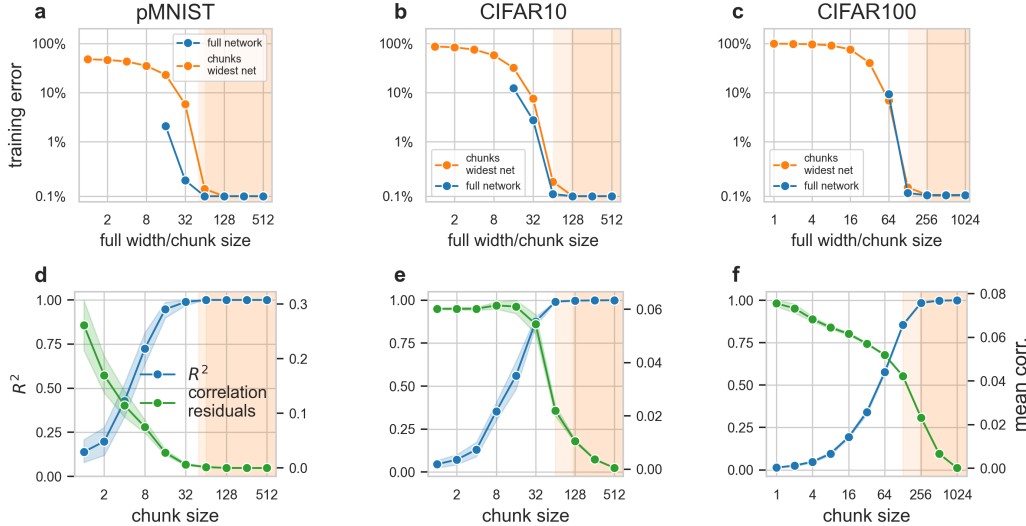

Figure 4: **The three signatures of representation redundancy. (i)** The training errors of the full networks (blue) and of the chunks taken from the widest network (orange) approach zero beyond a critical width/chunk size, resp. (panels **a-c**). **(ii)** The final representation of the widest network can be reconstructed from a chunk using linear regression (1) with an explained variance $R^2$ close to 1 (blue lines in panels **d-f**). **(iii)** The residuals of the linear map can be modeled as independent noise: we show this by plotting the mean correlation of these residuals (green line, panels **d-f**), averaged over 100 reconstructions starting from different chunks. A low correlation at high $R^2$ indicates that the chunk contains the information of the full representation with some statistically independent noise. *Experimental setup:* FC5 on pMNIST, Wide ResNet-28 on CIFAR10/100. Full details in Methods section 2.1

described as statistically independent random noise. The green profiles of the bottom panels of Fig. 4 show the *mean correlation* of the residuals of the linear fit (see Sec. 2.2). Below $w_c^*$, the residuals are not only large but also significantly correlated, since they are related to relevant features of the data that are not covered by the neurons of the chunk. As the chunk width increases above $w_c^*$, the correlation between residuals drops basically to zero. Therefore, in networks wider than $w_c^*$ any two chunks of equal size $w_c > w_c^*$ can be effectively considered as equivalent copies, or clones, of the same representation (that of the full layer), differing only by a small and non-correlated noise, consistently with the scaling law of the error shown in Fig. 3.

**The dynamics of training.** In the previous paragraphs, we set forth evidence in support of the hypothesis that large chunks of the final representation of wide DNNs behave approximately like an ensemble of independent measures of the full feature space. This allowed us to interpret the decay of the test error of the full networks with the network width observed empirically in Fig. 2. The three conditions that a chunked model satisfies in the regime in which its test error decays as $w_c^{-1/2}$ are represented in Fig. 4: (i) the training error of the chunked model is close to zero; (ii) the chunked model can be used to reconstruct the full final representation with an $R^2 \sim 1$ and (iii) the residuals of this reconstruction can be modeled as independent random noise. These three conditions are all observed at the end of the training. We now analyze the training *dynamics*. We will see that for the clones to arise, models not only need to be wide enough but also, crucially, they need to be trained to maximize their performance.

Clones are formed in two stages, which occur at different times during training. The first phase begins as soon as training starts: the network gradually adjusts the chunk representations in order to produce independent copies of the data manifold. This can be clearly observed in Fig. 5-a, which depicts the mean correlation between the residuals of the linear fit from the chunked to the full final representations of the network, the same quantity that we analyze in Fig. 4-(d-e-f, green profiles),

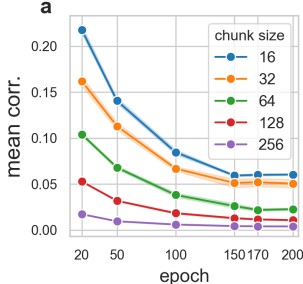
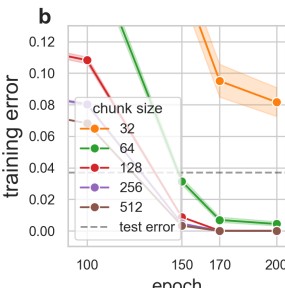
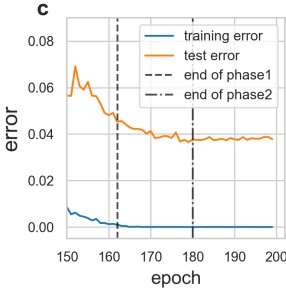

Figure 5: **The onset of clones during training. a:** As in Fig. 4, we show the mean correlation of the residuals of the linear reconstruction of the final representation from chunks, but this time as a function of training epochs. A small correlation indicates that the reconstruction error in going from chunks to final representation can be modeled as independent noise. Data obtained from the same WR28-8 trained on CIFAR10 as in Fig. 4. **b:** Training error during training for chunks of different sizes. After the network has reached zero training error at $\sim 160$ epochs, continuing to train improves the training accuracy of the chunks. **c:** Test and training error during training for the full network. Between epochs 160 and 180, the clones of the full network progressively achieve zero training error. In the same epochs, one observes a small improvement in the test error.

but now as a function of the training epoch. Both Figs. 4 and 5 analyze the WR28-8 on CIFAR10. As training proceeds, the correlations between residuals diminish gradually until epoch 160 and become particularly low for chunks greater than 64. After epoch 160 further training does not bring any sizeable reduction in their correlation. At epoch 160 the full network also achieves zero error on the training set, as shown in Fig. 5-b (brown) and Fig. 5-c (blue). This event marks the end of the first phase and the beginning of the second phase where the training error of the clones keeps decreasing while the full representation (blue) has already reached zero training error. For example, chunks of size 64 at epoch 150 have training errors comparable to the test error (dashed line of the middle panel). In the subsequent $\sim 20$ epochs the training error of clones of size 128 and 256 reaches exactly zero, and the training error of chunks of size 64 reaches a plateau.

Importantly, both phases improve the generalization properties of the network. This can be seen in Fig. 5-c, which reports the training and test error of the network, with the two phases highlighted. The figure shows that both phases lead to a reduction in the test error, although the first phase leads by far to the greatest reduction, consistent with the fact that the greatest improvements in accuracy typically arise during the first epochs of training. The formation of clones can be considered finished around epoch 180 when all the clones have reached almost zero error on the training set. After epoch 180 we also observe that the test error stops improving. In the Appendix (Sec. B) we report the same analysis done on CIFAR100 (see Fig. S1) and CIFAR10 trained on a DenseNet40 (see Fig. S2-(d-e-f)).

**Clones appear only in regularized networks.** So far in this work, we have shown only examples of regularized networks and data sets in which representations are redundant. However, if the network is not regularized, some of the signatures described above don't appear even if the width of the final representation is much larger than $W^*$ (the minimum interpolating width). Figure 6 shows the case of the Wide-ResNet28-8 analyzed in Fig. 5 trained on exactly the same data set (CIFAR10) but without weight decay. As shown in Fig.6-a in the network trained without regularization (blue line) the error does not scale as $w_c^{-1/2}$. This, as we have seen, indicates that the last hidden representation cannot be split in clones equivalent to the full layer. Indeed, the mean correlation of the residuals of the linear map of the chunks to the full representation remains approximately constant during training (Fig. 6-b), and is always much higher than what we observed for the same architecture and data set when training is performed with weight decay. We performed a similar analysis on the DenseNet40 (see Fig. S3), observing an analogous trend.

**Clones appear only if a network interpolates the training set: the case of ImageNet.** We saw that a chunk of neurons can be considered a clone if it fully captures the relevant features of the data, achieving almost zero training error (see Fig. 4). This condition is not satisfied for most of the

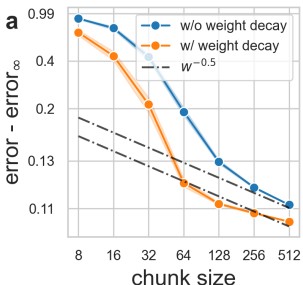 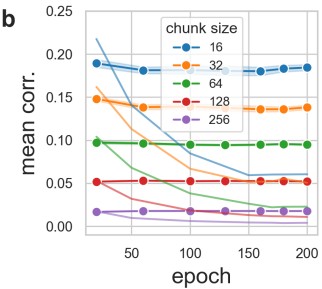

Figure 6: **A network trained without weight decay on CIFAR10. a:** the test error of chunks of a Wide-ResNet28-8 trained without weight decay (blue) and with weight decay (orange, taken from Figure 3-b). **b:** Mean correlation between residuals of the linear reconstruction of the full representation from chunks of different sizes for two networks: one trained without weight decay (thick lines), and one using weight decay (thin lines, same data as in Fig. 5-a).

networks trained on ImageNet [16], therefore we do not expect to see redundant representations in this important case. We verified this hypothesis by training a family of ResNet50s where we multiply all the channels of the layers after the input stem by a constant factor $c \in \{0.25, 0.5, 1, 2, 4\}$. In this manner the widest final representation we consider consists of $8192$ neurons, which is four times wider than both the standard ResNet50 [29] and its wider version [26] (see Sec. 2.1). We trained all the networks following the standard protocols and achieved test errors comparable to or slightly lower than those reported in the literature (see Appendix, Sec. A). We find that even in the case of the largest ResNet50, the top-1 error on the training set is $\sim 8\%$ (see Fig. 7-a) and the network does not achieve interpolation, as discussed also in [16].

In this setting, none of the elements associated with the development of independent clones can be observed. The scaling of the test error of the chunks is steeper than $w_c^{-1/2}$ (see Fig. 7-b) suggesting that chunks remain significantly correlated to each other. Figure 7-c shows that the mean correlation of the residuals does not decrease during training, as it happens for the networks we trained on CIFAR10 and CIFAR100. We conclude that in a ResNet50, a representation with 8192 neurons is too narrow to encode all the relevant features redundantly on ImageNet, and a chunk as large as 4096 activations is not able to reconstruct all the relevant variations of the data as it does in the cases analyzed in Sec. 3.

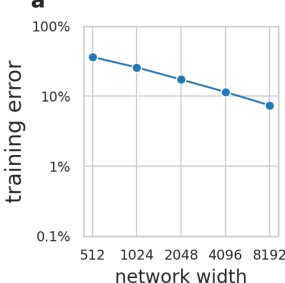 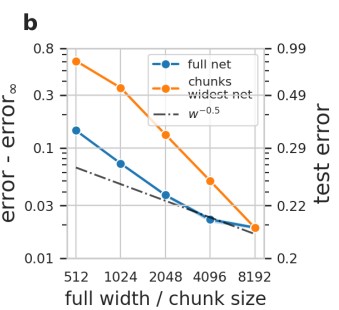 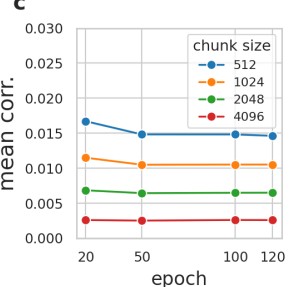

Figure 7: **ResNet50 trained on ImageNet a:** ImageNet training error as a function of the ResNet50 width. **b:** Decay of the test error as a function of the network width (blue) and for chunks of the widest ResNet50 (orange) to the error of an ensemble of ResNet50-4. The ensemble consists of four networks. **c:** Mean correlation (see Sec. 2.2) of the residuals of the linear map of a chunk of the last hidden representation to the full representation. The network analyzed is ResNet50-4.

# 4  Discussion

This work is an attempt to explain the paradoxical observation that over-parameterization boosts the performance of DNNs. This "paradox" is not a peculiarity of DNNs: if one trains a prediction model with $n$ parameters using the same training set, but starting from independent initial weights and receiving samples in an independent way, one can obtain, say, $m$ models which, in suitable conditions, provide predictions of the same quantity with independent noise due to initialization, SGD schedule, etc. If one estimates the target quantity by an ensemble average, the statistical error will (ideally) scale with $m^{-1/2}$, and therefore with $N^{-1/2}$, where $N = n\,m$ is the total number of parameters of the combined model. This will happen even if $N$ is much larger than the number of data.

What is less trivial is that a DNN can accomplish this scaling within a single model, in which all the parameters are optimized collectively via the minimization of a single loss function. Our work describes a possible mechanism at the basis of this phenomenon in the special case of neural networks in which the last layer is very wide and the model is regularized. We observe that if the layer is wide enough, random subsets of its neurons can be viewed as approximately independent representations of the same data manifold (or clones). This implies a scaling of the error with the width of the layer as $W^{-1/2}$, which is qualitatively consistent with our observations.

**The impact of network architecture.**   The capability of a network to produce statistically independent clones is a genuine effect of the over-parametrization of the *whole* network as we find that redundancies appear even if the last layer width is kept constant and the width of all intermediate layers is increased (see Appendix, Sec. B, Fig. S4-a, ). At the same time, we also verified that if the network is too narrow to interpolate the training set, increasing the width of *only* the final representation is not sufficient to make the last layer redundant. We give an example of this effect in Fig. S4-b, where we show that the test error of a WR28-1 on CIFAR10 does not decrease if only the width of the final representation is increased, while the rest of the architecture is kept at a constant width.

**The impact of training.**   The mechanism we described is robust to different training objectives since we trained the convolutional networks with cross-entropy loss and the fully connected networks with hinge loss. However, even for wide enough architectures, clones appear only if the training is continued until the training error reaches zero. In our examples, by stopping the training too early, for example when the training error is similar to the test error, the chunks of the last representation would not become entirely independent from one another, and therefore they could not be considered clones.

**Neural scaling laws.**   Capturing the asymptotic performance of neural networks via scaling laws is an active research area. Hestness *et al.* [33] gave an experimental analysis of scaling laws w.r.t. the training data set size in a variety of domains. Rosenfeld *et al.* and Kaplan *et al.* [34, 35] experimentally explored the scaling of the generalization error of deep networks with the number of parameters/data points across architectures and application domains for supervised learning, while Henighan *et al.* [36] identified empirical scaling laws in generative models. Bahri *et al.* [37] showed the existence of four scaling regimes and described them theoretically in the NTK or *lazy regime* [38–40], where the network weights stay close to their initial values throughout training. None of these works propose a mechanism that would explain these scalings with properties of the representation. Geiger *et al.* found that the generalization error can be related to the fluctuations of the output induced by initialization and showed that it scales as $W^{-1}$ in networks trained *without* weight decay both in the NTK [21] and in the mean field [41] regimes. We instead consider the feature learning regime and train our networks with weight decay which is unavoidable to obtain models with state-of-the-art performance. This might explain the difference in the scaling law that we observe empirically. Previous theoretical work did not study the impact of weight decay on scaling laws, so we hope that our results can spark further studies on the role of this essential regularizer.

**Relation to theoretical results in the mean-field regime.**   Our empirical results also agree with recent theoretical results that were obtained for two-layer neural networks [42–47]. These works characterize the optimal solutions of two-layer networks trained on synthetic data sets with some controlled features. In the limit of infinite training data, these optimal solutions correspond to

networks where neurons in the hidden layer duplicate the key features of the data. These "denoising solutions" or "distributional fixed points" were found for networks with wide hidden layers [42–45] and wide input dimension [46, 47]. Another point of connection with the theoretical literature is the concept of *dropout stability*. A network is said to be $\epsilon$-dropout stable if its training loss changes by less than $\epsilon$ when half the neurons are removed at random from each of its layers [48]. Dropout stability has been rigorously linked to several phenomena in neural networks, such as the connectedness of the minima of their training landscape [49, 50].

**Bias-variance trade-off and implicit ensembling**  The success of various deep learning architectures and techniques has been linked to some form of ensembling. The successful dropout regularisation technique [51, 52] samples from an exponential number of "thinned" networks during training to prevent co-adaptation of hidden units. While this can be seen as a form of (implicit) ensembling, here we observe that co-adaptation of hidden units in the form of clones occurs *without* dropout, and is crucial for their improving performance with width. Recent theoretical work on random features showed that ensembling and over-parameterization are two sides of the same coin and that both mitigate the increase in the variance of the network that classically leads to *worse* performance with over-parameterization due to the bias-variance trade-off [18–20]. The plots of bias and variance in Fig. S5 for the architectures trained on the CIFAR10 and CIFAR100 data sets show that the clone size in these cases is slightly above the peak of the variance and almost coincides with the interpolation width of the full networks of the same size.

**Impact for applications.** The framework introduced in this work allows verifying if a neural network is sufficiently expressive to encode multiple statistically independent representations of the same ground truth, which, we believe, is a fair proxy of model quality and robustness. In particular, we find that reaching interpolation on the training set is not necessarily detrimental for generalization, and is instead a necessary condition for developing redundancies which, in turn, reduces the test error.

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
