# A  Hyperparameters used and training procedures

**Fully-connected networks on MNIST.**  We train the fully-connected networks for 5000 epochs with stochastic gradient descent using the following hyperparameters: batch size = 256, momentum = 0.9, learning rate = $10^{-3}$, weight decay = $10^{-2}$. We optimize our networks using Adam.

**Wide-ResNet-28 and DenseNet40-BC on CIFAR10/100.**  All the models are trained for 200 epochs with stochastic gradient descent with a batch size = 128, momentum = 0.9, and cosine annealing scheduler starting with a learning rate of 0.1. The training set is augmented with horizontal flips with 50% probability and random cropping the images padded with four pixels on each side. On CIFAR10 trained on WR28 we select a weight decay equal to $5 \cdot 10^{-4}$ and label smoothing magnitudes equal to 0.1 for WR28-{0.25, 0.5, 1, 2} and equal to 0 for WR28-{4, 8}. On CIFAR10 trained on Densenet40-BC we set a weight decay equal to $5 \cdot 10^{-4}$ and label smoothing magnitudes equal to 0.05 for all the networks On CIFAR100 trained on WR28 we set weight decays equal to {10, 7, 5, 5, 5}$\cdot 10^{-4}$ and label smoothing magnitudes equal to {0.1, 0.07, 0.05, 0, 0} for WR28-{1, 2, 4, 8, 16} respectively. All the hyperparameters were selected with a small grid search.

**ResNet50 on ImageNet.**  We train all the ResNet50 with mixed precision [53] for 120 epochs with a weight decay of $4 \cdot 10^{-5}$ and label smoothing rate of 0.1 [54]. The input size is $224 \times 224$ and the training set is augmented with random crops and horizontal flips with 50% probability. The per-GPU batch size is set to 128 and is halved for the widest networks to fit in the GPU memory. The networks are trained on 8 or 16 Volta V100 GPUs so as to keep the batch size $B$ equal to 1024. The learning rate is increased linearly from 0 to $0.1 \cdot B/256$ [55] for the first five epochs and then annealed to zero with a cosine schedule.

Table 1: Test accuracy (average over four runs)

| CIFAR10 | | CIFAR100 | | ImageNet (top1) | |
|---|---|---|---|---|---|
| network | accuracy | network | accuracy | network | accuracy |
| Wide-RN28-0.25 | 84.1 | Wide-RN28-1 | 70.4 | RN50-0.25 | 67.0 |
| Wide-RN28-0.5 | 90.3 | Wide-RN28-2 | 75.7 | RN50-0.5 | 74.1 |
| Wide-RN28-1 | 93.4 | Wide-RN28-4 | 79.6 | RN50-1 | 77.6 |
| Wide-RN28-2 | 95.2 | Wide-RN28-8 | 80.8 | RN50-2 | 79.1 |
| Wide-RN28-4 | 95.9 | Wide-RN28-16 | 81.9 | RN50-4 | 79.5 |
| Wide-RN28-8 | 96.1 | | | | |
| DenseNet40-BC (k=8) | 91.6 | | | | |
| DenseNet40-BC (k=16) | 93.9 | | | | |
| DenseNet40-BC (k=32) | 95.1 | | | | |
| DenseNet40-BC (k=64) | 95.7 | | | | |
| DenseNet40-BC (k=128) | 96.0 | | | | |

# B  Additional experiments

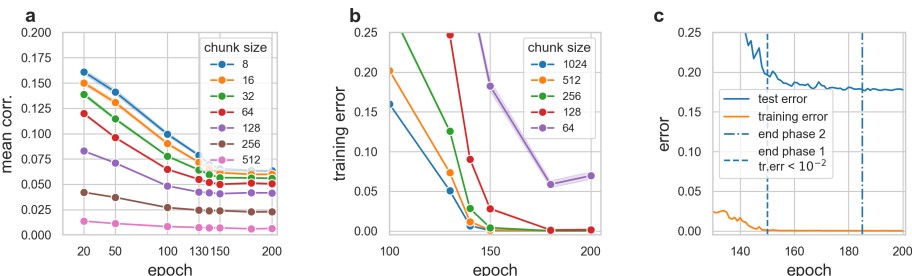

Figure S1: **Training dynamics on CIFAR100. a:** As in Fig. 4, we show the mean correlation of the residuals of the linear reconstruction of the final representation of a Wide-Resnet28-8 from chunks, but this time as a function of training epochs. A small correlation indicates that the reconstruction error in going from chunks to the final representation can be modeled as independent noise. **b:** Training error of chunks of a Wide-Resnet28-8 and its full layer representation. From epoch 150 to epoch 185 the training error of the chunks with size 128/256 decreases below 0.5%, while for smaller chunk sizes it remains above 5%. Random chunks with sizes larger than 128/256 can fit the training set, thus having the same representational power as the whole network on the training data. For W > 128/256 the test accuracy is decaying approximately with the same law as that of independent networks with the same width (see Fig. 3). This picture suggests that for CIFAR100 the size of a clone is 128/256, slightly larger than the size of the clones in CIFAR10. **c:** Training and test error dynamics for the same Wide-ResNet28-8. After epoch 150 the training error of the full network remains consistently smaller than 0.1% (orange profile) while the test error continues to decrease until epoch 185 from 0.194 to 0.1765 (blue profile). In the same range of epochs (150-185) the training error of smaller chunks decreases sensibly (see panel **b**).

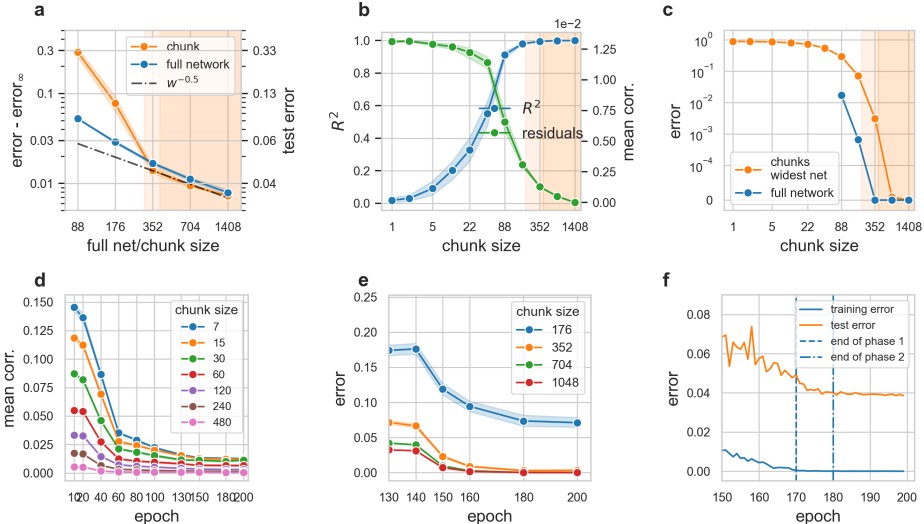

Figure S2: **A DenseNet40 architecture. a:** Decay of the test error of independent networks (blue) and chunks of the widest network (orange) to the error of an ensemble average of ten of the widest networks (DenseNet40-BC, k=128) **b:** Blue profile: $R^2$ coefficient of the ridge regression of a chunk of $w_c$ neurons ($x$-axis) to the full layer representation. Green profile: mean correlation of the residuals of the mapping as described in Sec. 2.2. **c:** Training error of various DenseNet40 of increasing width (blue) and of chunks of the widest architecture (orange). **d:** The mean correlation of the residuals from the linear reconstruction of the final representation from chunks of a given size for a DenseNet40-BC (k=128) during training. **e:** Training error dynamics of chunks of a DenseNet40-BC (k=128). **f:** Training and test error dynamics for a DenseNet40-BC (k=128).

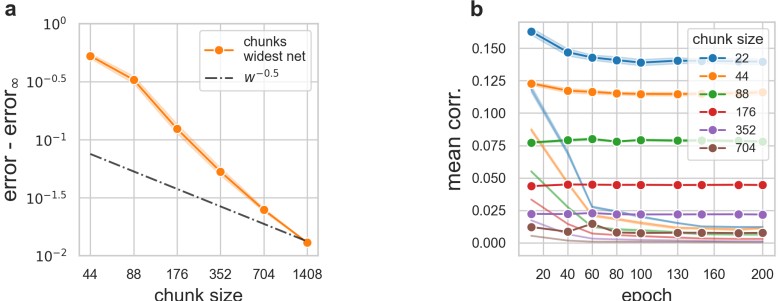

Figure S3: **A Densenet40 not regularized.** A DenseNet40-BC (k=128) trained on CIFAR10 without weight decay. This experiment reproduces on a DenseNet the analysis shown on a Wide-ResNet28 in Sec. 3. It shows that **a**: also in a DenseNet architecture not well regularized error -error$_\infty$ decays faster than $w_c^{-1/2}$ and **b**: the mean correlation of the residuals do not decrease during training. The thin profiles of panel **b** are the same as those shown in Fig. S2-d.

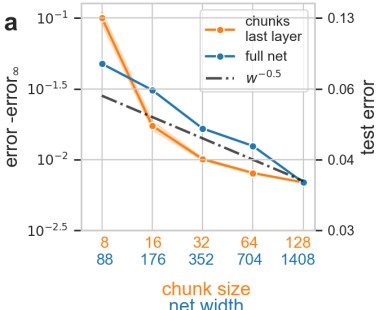 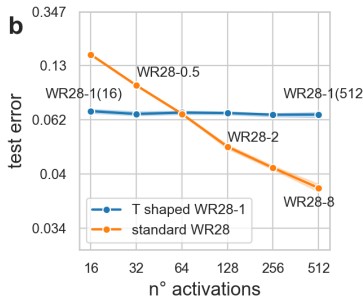

Figure S4: **Impact of the width of the intermediate layers.** We study how the scaling of the test error is affected (Fig. **a**) by increasing the width of the intermediate representations while keeping the width of the last layer constant or (Fig. **b**) by increasing the last layer width while keeping the width of the network constant. In S4-**a** we trained DenseNet40 on CIFAR10 with an additional $1 \times 1$ convolution to keep the number of output channels fixed at 128. Figure S4-**a** shows that increasing the width of intermediate layers makes the test accuracy of the full network decay approximately as $w_c^{-1/2}$, even when the width of the final representation is fixed. A bottleneck of 128 channels makes the clones much smaller: The orange profile shows that a strong deviation from the $w^{-1/2}$ can be seen for chunk sizes smaller than 16 (vs 350 Fig. 1b, main paper). We also verified that 16 random neurons are sufficient to interpolate the training set (error $< 5 \cdot 10^{-3}$) and that the $R^2$ coefficient of fit to the full layer is 0.912 (0.98 for chunk sizes = 32). The phenomenology described in the manuscript applies also when a bottleneck of 128 channels is added at the end of the network. In a second experiment, we trained a ResNet28-1 increasing only the number of channels in the last layer. We modified the number of output channels of the last block of conv4 and analyzed the representation after average pooling, as we did in the other experiments. The network was trained for 200 epochs using the same hyperparameters and protocol described in Sec. 2. Figure S4-**b** shows that the test error of the modified ResNet28-1 is approximately constant (blue profile). On the contrary, when we increase the width of the whole network the test error decays to the asymptotic test error with an approximate scaling of $1/\sqrt{w}$ (orange profile).

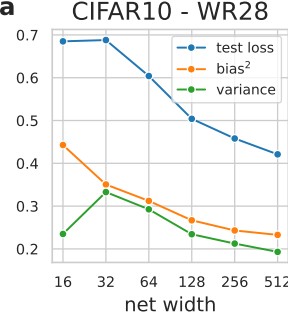
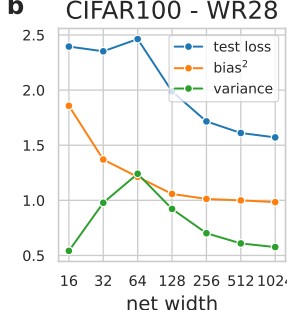
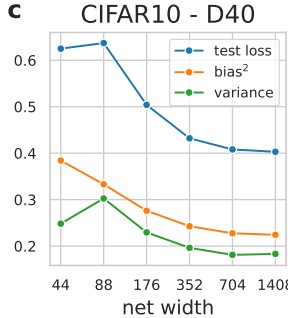

Figure S5: **Bias-variance profiles in CIFRA10 and CIFAR100.** We compute the bias and the variance profiles for the convolutional architectures analyzed in the paper: Wide-ResNets and DenseNets trained on CIFAR10, and Wide-ResNets trained on CIFAR100. Since we trained the models using the cross-entropy loss, the standard bias-variance decomposition, which assumes the square loss, does not apply. Instead, we used the method recently proposed by Yang *et al.* [56] to estimate the bias and the variance on networks trained with cross-entropy loss. The average over the data distribution is approximated by splitting the CIFAR training sets into five disjoint subsets containing 10 000 images each and training the networks from scratch on each of them. We use the same regularization for all the networks, namely that of the largest architectures, with weight decay equal to $5 \cdot 10^{-4}$ and label smoothing equal to 0. We repeat the procedure 4 times, for a total of 20 training runs for each network width, as described in Ref. [56]. We show the test loss curves as well as the squared bias and variance. As expected, the bias of the models decreases as we add parameters and make the model more flexible. The variance of the models initially grows with width to reach its peak at $W_{\mathrm{peak}} = 32$ and $64$ for CIFAR10 and CIFAR100 trained on Wide-ResNet28 (a, b) and $W_{\mathrm{peak}} = 88$ on CIFAR10 trained on DenseNet40 (c). As we increase the width, the variance decreases, allowing the model to generalize better and better and defying the classical bias-variance trade-off. The clone size $w_c^*$ for these architectures are slightly above the widths at which the variance peaks and are $w_c^* = 64$, and $128$ for CIFAR10 and CIFAR100 trained on Wide-ResNet28 (compare Figs. 3 and 4) and $w_c^* = 170/250$ (Fig. S2). In all cases, the onset of the clones occurs at a width that is approximately two times larger than $W_{\mathrm{peak}}$, similar to the width at which an architecture of size $w_c^*$ interpolates the training set.