# OpenReview forum: "Redundant representations help generalization in wide neural networks"
_NeurIPS.cc/2022/Conference — NeurIPS 2022 Accept_

### Official Review · Reviewer_Ef6D · 2022-07-07

**Rating:** 5
**Confidence:** 4
**Soundness:** 3 good
**Presentation:** 2 fair
**Contribution:** 1 poor

**Summary:**

This paper studies the representation of the last hidden layer of wide neural networks. The authors observe that these representations for networks that achieve zero training error contain redundant "clones". It is observed that by randomly dropping neurons of the last hidden layer, the performance is similar to training the network in the reduced size. This could serve as an explanation for the benign overfitting of over-parameterized neural networks.

**Questions:**

1- If redundant representations help generalization for wide enough neural networks, then we should observe that for the full network in Figure 1 (b), the error is lower than the chunk network. However, for large widths, we are observing the opposite. Then, either the explanation made in the introduction is not clear or the conclusion simply doesn't hold.

2- Figure 1: Why the reference for error is error_inf? Why not report the error itself? Also, what is the classification error computed on?

3- In finding number 1, what do you mean by "w_c predicts the output with an accuracy that scales with w_c^(-0.5)"? Why is the slope of the figure of error versus net size referred to as "accuracy"?

4- Why is w_c selected randomly? Why not select them from the correlation values to ensure the correlated neurons are not selected together and only the uncorrelated ones are selected together?

5- The main motivation behind the work is its explainability of the bias-variance mystery. However, the exact connection has not been made. How would the bias-variance curve look in your experiments?

6- Does the point regarding regularization only apply to weight decay or any form of regularization? If the latter, then the scope of the finding would be very limited. Also, what is the explanation behind this?


**Limitations:**

Yes, the authors mention the limitation of the observation in finding number 3.

**Strengths And Weaknesses:**

Strengths:

1- The observation is novel and interesting.

2- Some ablation studies including studying the intermediate layers have been reported.

Weaknesses:

1- A missing discussion from the paper is the connection between the presented observation and pruning neural networks. If it's the case that neurons are redundant, how can we select the redundant ones and drop them to have pruned networks? This could be a useful pruning strategy that is not covered in the paper. Overall, the importance of the outcome of the study is not clear and no application is shown.

2- The significance of the result is limited: The scope of the observation is limited to having zero training error and using weight decay. If the network is stopped early or there is no regularization the observations won't hold. Also, the last layer of the neural network should be very wide.

---

> ### Author Response · Authors · 2022-07-30
> **Response to reviewer Ef6D**
>
> We thank you for your time reviewing the manuscript and for your detailed comments and questions. We reply to both point-by-point below, and we report an additional experiment following your suggestion to investigate the bias-variance trade-off of our models. We hope that the replies will address your concerns. If so, we would appreciate it if you adjusted your score; if not, please let us know. Thank you!
>
>
> > 1. Connection to pruning
>
> Pruning aims to decrease the size of the network *without* deteriorating performance, which is usually achieved by retraining. Conversely, we show that decreasing the size of the last layer by removing random subsets of activations *does* decrease performance, meaning that our approach cannot be used as an alternative pruning method. However, we do not retrain the weights in our experiments. We can say that our work provides an additional theoretical foundation for pruning, since we provide extensive evidence of the high redundancy present in wide neural networks, which might be one of the features that make pruning effective.
>
> > 2. The significance of the result is limited: The scope of the observation is limited to having zero training error and using weight decay.
>
> The key result of our work is to identify a mechanism that allows deep convolutional neural networks to fit their training data perfectly *and* to achieve state-of-the-art generalisation performance at the same time. Hence having zero training error and using weight decay are not limitations of our study; they are simply characteristics of the best-performing models on CIFAR10/100 that we aim to study.
>
> Indeed, when training the models, we simply followed common practice to set the hyperparameters and training protocols (cf. Refs. [3-11] below). The accuracies we report in Table 1 are consistent with those reported in these papers, which are considered the state of the art on CIFAR10/100 datasets on these architectures. The protocols used in **all these papers** naturally lead to **zero training error** on CIFAR10/100 at the end of training, also in models of moderate size, without us enforcing it in any manner.
>
> Similarly, weight decay is part of all the protocols from the papers cited below. Most of them choose a value equal to 0.0005, as we do. We also use the same starting learning rate of 0.1, the same data augmentation and the same cosine annealing scheduler found in many of them. We did our best to follow the common wisdom of the papers that introduce these models.
>
> > 2. [continued] If the network is stopped early or there is no regularization the observations won't hold.
>
> We *do* use early stopping, and plot all our results with the model that performed best on the test set during training, again to mimic what is normally done when training such models in practice (see e.g. [11]). Specifically, the results shown in figures 2-4 for CIFAR10 on Wide-ResNet28 are done with the model of epoch 180 (see fig. 5c); the same holds for all the other data sets and architectures.
>
> > 2. [continued] Also, the last layer of the neural network should be very wide.
>
> The architectures we study are chosen to represent the shape and sizes of the models commonly used in practical applications. We mention that the size of the standard benchmark model used to validate novel training techniques is Wide-Resnet28-10 [8-10]. Our wider ResNet version (Wide-Resnet28-16 on CIFAR100) is only marginally larger.
>
> ### Questions
>
> > Q1. If redundant representations help generalization for wide enough neural networks, then we should observe that for the full network in Figure 1 (b), the error is lower than the chunk network. However, for large widths, we are observing the opposite. Then, either the explanation made in the introduction is not clear or the conclusion simply doesn't hold.
>
> The error for the chunks in Fig. 1b is lower than that of a network of the same width trained end-to-end because the chunks are taken from the largest network (cf. lines 31-33). Hence the chunks are part of the final representation of a much larger network that extracted better features, as shown by its lower generalization accuracy (blue line in Fig 1b).
>
> > Q2. Figure 1: Why the reference for error is error_inf? Why not report the error itself? Also, what is the classification error computed on?
>
> Since we are training on a finite data set, even the best models will have a finite error as their width goes to infinity. We estimate this error at "width infinity" by  taking an ensemble average of the widest network; scaling laws for the decay of the error can then be observed by analyzing how the error of a single network approaches the error at "width infinity". This procedure to disentangle the effect of finite training set size from network width is commonly performed in the literature (cf. Geiger et al. [21] and Bahri et al. [37])
>
> Please also notice that the  vanilla classification error is shown in Figure 3.

---

> > ### Author Response · Authors · 2022-07-30
> > **Continuation response to reviewer Ef6D**
> >
> > > Q3. In finding number 1, what do you mean by "w_c predicts the output with an accuracy that scales with w_c^(-0.5)"? Why is the slope of the figure of error versus net size referred to as "accuracy"?
> >
> > The referee is right, "accuracy" should be changed to "error".
> >
> > > Q4. Why is w_c selected randomly? Why not select them from the correlation values to ensure the correlated neurons are not selected together and only the uncorrelated ones are selected together?
> >
> > This is an interesting suggestion. Selecting the w_c neurons with a specific protocol, for example by forming a clone out of neurons with low correlation as the reviewer suggests, may indeed allow finding smaller subsets of neurons that reproduce the full representation. This procedure would presumable also lead to a faster decay of the error of the chunks with chunk size, since the neurons are not independently selected anymore. This procedure would not change the number of clones contained in a representation though, it would only change their size: one would end up with a partitioning of the representation in clones which excludes part of the neurons, those which do not bring a statically independent signal.
> >
> > > Q5. The main motivation behind the work is its explainability of the bias-variance mystery. However, the exact connection has not been made. How would the bias-variance curve look in your experiments?
> >
> > The main motivation of our work is to explain why adding parameters to a DNN that has reached zero training error will typically improve its generalisation performance. While this is of course connected to the bias-variance trade-off, analysing the bias and the variance of our models was not our focus. Following the reviewer's suggestion, we did however compute the bias and the variance of a set of DenseNet40s with increasing width, trained on CIFAR10. To approximate the average over the data distribution, we trained each DenseNet40 from scratch on six different subsets of CIFAR10, each containing 20k samples. Since we trained the models using the cross-entropy loss, the standard bias-variance decomposition, which assumes the square loss, does not apply. Instead, we used the method recently proposed by Yang et al. [12] to estimate the bias and the variance of the DenseNets40.
> >
> > We show the error curves of the models as well as their bias and variance in Figure "response_rev_figures/2_bias_variance.pdf", which we uploaded to our anonymous repository: https://anonymous.4open.science/r/redundant_representation2022-E4FD.  As expected classically, the bias of the model decreases as we add parameters and make the model more flexible. The variance of the models initially increases with width, as in classical statistical learning models. However, as we increase the width even more, the variance starts to decrease, allowing the model to generalise better and better, defying the classical bias-variance trade-off. The onset of the non-classical regime approximately coincides with the appearance of the clones (as shown by the value of the R^2 coefficient, which goes to 1 exactly for clones of size 88). The decrease of the variance is therefore connected with the redundancy of the representations, as measured by the R^2.
> >
> > We will mention the results of this analysis in the discussion, adding a figure in the appendix.
> >
> > > Q6. Does the point regarding regularization only apply to weight decay or any form of regularization? If the latter, then the scope of the finding would be very limited. Also, what is the explanation behind this?
> >
> > Weight decay is essential to observe cloning. This regularization procedure is  equivalent  an $\ell_2$ regularization, or to adding a Gaussian prior on the value of the weights. Therefore our setup explicitly or implicitly covers most of the regularization procedures which are used in deep learning, but not all (for example, not data augmentation or Dropout or early stopping used on its own).
> >
> > The explanation behind this remains a challenging theoretical problem as seen by the fact that most theoretical works tried to explain the scaling law of generalization error in the neural tangent kernel regime **without weight decay** see Geiger et al. [21] and Bahri et al. [37] (main paper) and Dyer & Gur-Ari [13] (below).
> >
> > We hope that our results can spark further studies on the role of this essential regularizer, and we think that clearly establishing a list of conditions which should be satisfied in order to observe a phenomenon, in this case benign overfitting, is not a limitation of our work, but instead its value, especially as these conditions are compliant with the state of the art on deep learning.

---

> > > ### Author Response · Authors · 2022-07-30
> > > **References**
> > >
> > > [1] Neyshabur et al. In search of the real inductive bias: on the role of implicit regularization in deep learning. ICLR workshop 2015. [arXiv:1412.6614](https://arxiv.org/abs/1412.6614)
> > >
> > > [2] Zhang et al. Understanding deep learning requires rethinking generalization. ICLR 2017 [arXiv:1611.03530](https://arxiv.org/abs/1611.03530)
> > >
> > > [3] Loshchilov & Hutter. SGDR: Stochastic Gradient Descent with Warm Restarts. ICLR 2017. [openReview](https://openreview.net/forum?id=Skq89Scxx)
> > >
> > > [4] Zagoruyko & Komodakis: Wide Residual Networks. [arXiv:1605.07146](https://arxiv.org/abs/1605.07146)
> > >
> > > [5] Bello et al. Revisiting ResNets: Improved Training and Scaling Strategies. [arXiv:2103.07579](https://arxiv.org/abs/2103.07579)
> > >
> > > [6] Cubuk et al. AutoAugment: Learning Augmentation Policies from Data. CVPR 2019. [arXiv:1805.09501](https://arxiv.org/abs/1805.09501)
> > >
> > > [7] Hu et al. Squeeze-and-Excitation Networks. CVPR 2018. [arXiv:1709.01507](https://arxiv.org/abs/1709.01507)
> > >
> > > [8] Foret et al. Sharpness-Aware Minimization for Efficiently Improving Generalization. ICLR 2021. [arXiv:2010.01412](https://arxiv.org/abs/2010.01412)
> > >
> > > [9] Cubuk et al. RandAugment: Practical Automated Data Augmentation with a Reduced Search Space. [NeurIPS '20](https://proceedings.neurips.cc/paper/2020/hash/d85b63ef0ccb114d0a3bb7b7d808028f-Abstract.html)
> > >
> > > [10] Han, Kim & Kim: Deep Pyramidal Residual Networks. CVPR 2017. [arXiv:1610.02915](https://arxiv.org/abs/1610.02915)
> > >
> > > [11] Huang et al. Densely Connected Convolutional Networks. CVPR 2017. [arXiv:1608.06993](https://arxiv.org/abs/1608.06993)
> > >
> > > [12] Yang et al. Rethinking Bias-Variance Trade-off for Generalization of Neural Networks. ICML 2020. [https://arxiv.org/abs/2002.11328](arXiv:2002.11328)
> > >
> > > [13] Dyer & Gur-Ari Asymptotics of Wide Networks from Feynman Diagrams. ICLR 2020
> > > [openreview](https://openreview.net/forum?id=S1gFvANKDS)

---

### Official Review · Reviewer_NAgd · 2022-07-10

**Rating:** 6
**Confidence:** 3
**Soundness:** 3 good
**Presentation:** 2 fair
**Contribution:** 2 fair

**Summary:**

This paper studied the statistical properties of randomly selected chunks of neurons in the last layer representations of wide deep neural networks.

Specifically, the authors observed that when the chunk size is larger than a threshold, the test error is similar to that of a full network of the same width under a decay rate of $-1/2$. This implies that sufficiently large chunks can be viewed as statistically independent estimators of the same features of the data (referred to as "clone"), and the authors further characterize the various properties of "clones".

**Questions:**

Do the authors consider conducting their experiments on ImageNet with a model with higher redundancy, e.g., increase the width of the intermediate layers in addition to the final layer? It is interesting to see if clones exist and their properties.



**Ethics Review Area:**

["I don’t know"]

**Limitations:**

No limitations.

**Strengths And Weaknesses:**

Strengths:

1. The observation that sufficiently large chunks of neurons can be viewed as independent representations of the same data manifold is interesting. It may provide new insights for the connection between redundancy and generalization, model pruning and ensemble learning.
2. The hypothesis is well-motivated by observations on multiple architectures and datasets.
3. Extensive quantitative analysis has been conducted to verify the hypothesis and characterize the properties of "clones".

Weakness:

1. The study is only based on small scale datasets and networks. It seems that clones appear only when the model has sufficient capacity to interpolate the training data. As the properties of clones remain unknown in larger-scale settings, further implications and applications can be limited.

Minor:

1. Missing y-axis in Figure 1.b.

---

> ### Author Response · Authors · 2022-07-30
> **Response to reviewer NAgd**
>
> Thank you for your time reviewing the paper. We address your concerns and questions point-by-point below. We hope that we can clarify your concerns, if not, please let us know. Thank you!
>
> > The study is only based on small scale datasets and networks. It seems that clones appear only when the model has sufficient capacity to interpolate the training data.
>
> The reviewer is correct in that clones only form when the models are large enough to interpolate their training data, and indeed this is one of our key results. Our goal from the outset was to understand how interpolating classifiers are able to achieve state-of-the-art generalisation despite fitting all their training data, which is baffling from the point of view of the classical bias-variance trade-off.
>
> As we mention in a related answer to reviewer Ef6D, we based our analysis on the most cited and commonly used models in the recent literature. The largest models we analyze (e.g. DenseNet40\_128, Wide ResNet28\_16) are not so "small scale" as they almost fill the memory of a V100 GPU of 16GB during training.
>
> We trained them using the recipes found on those relevant papers and simply observe that most of these models, already at moderate sizes, achieve zero training error when trained on CIFAR data sets.
> Hence we do not consider the restriction of studying interpolating classifiers a limitation, because the standard architectures used in practice very often reach interpolation.
>
> This interpolation regime is instead not reached on ImageNet on the network we analyzed, because in this case, the models we have used not (yet) large enough. We can not afford to increase the width of a network deeper then ResNet50 on ImageNet due to the huge memory cost of these models.
>
> ### Minor:
>
> > Missing y-axis in Figure 1.b.
>
> Thank you, we added this.
>
> ### Questions
>
> > Do the authors consider conducting their experiments on ImageNet with a model with higher redundancy, e.g., increase the width of the intermediate layers in addition to the final layer?
>
> We increased the width of all the intermediate layers of the ResNet50s we analyzed, not only the width of the final layer (see l 82-83).  Indeed, in Fig. 11b we verified that increasing the width of the last layer alone while keeping the width of the rest of the network fixed, is not sufficient to improve the test accuracy and to observe clones even on the networks trained on CIFAR-datasets.

---

### Official Review · Reviewer_aMM9 · 2022-07-11

**Rating:** 7
**Confidence:** 3
**Soundness:** 3 good
**Presentation:** 3 good
**Contribution:** 4 excellent

**Summary:**

The paper provided an insight into the “benign overfitting” of convolutional neural networks, by inspecting the last hidden layer representations of the networks. They observed and proposed an explanation of the phenomenon that the neurons tend to split into groups carrying duplicated information when the network is wide enough and the training error is zero.

**Questions:**

1. Limitations on architectures: the research is focused on convolutional neural networks.
    - Does it generalize to Transformers or other networks with attention maps?
    - How much does it depend on the existence of the convolution units, which have limited span? A related question: by changing the filter size of the convolutional layers from small to very large, will the following claim hold “... neurons tend to split into groups that carry identical information”?

2. In several places of the work, e.g. Figure 5, the author(s) mentioned that the training error reached zero, and they continued the training to get the claimed results. A naive question: if the training error stays zero, where does the gradient come from?
    - Is the training error actually very small? If so, it is important to show the training curve or relevant statistics during the stage where the neurons are trying to duplicate themselves, in order to understand whether there is any phase transition or dynamical changes.
    - If the training error is actually zero, the updates of parameters may solely be due to weight decay. If so, the weights are simply shrinking by a factor of (1-learning_rate * weight_decay_factor) per step after the training error reaches 0, and this effect alone introduces the phenomenon of duplicated neurons. Then this would be a very useful training recipe, which could be validated and applied on broader sets of architectures.

3. It is claimed that “clones appear if the training error is zero and the model is trained with weight decay”. I understand that weight decay is a way of regularization. But is there any explanation why standard regularizations, e.g. imposing L2 norm or Gaussian priors, don’t work? Or if other regularizations also work, how do they compare with using weight decay?

4. The main claim of “distinct clones provide independent measures of the same salient features of the data” seems to rely on the fact that the error decays approximately as W^{-1/2}, when the chunk width W goes beyond critical chunk size w_c. However, in Figure 2, there are only 2 data points per model beyond w_c=256; in Figure 3, the W^{-1/2} law seems not so well for CIFAR 10 and CIFAR 100; in Figure 8, it seems far away from W^{-1/2} for ImageNet.
    - Based on these, apparently, the empirical claim of “the error of the chunks with w neurons decays as w^{-½}” is a weak claim, especially for more complex datasets CIFAR 100 and ImageNet. Are there any other experiments or insights to support the main claim?

5. Is there direct proof of the claim in line 152 “In the following we will indeed see that distinct clones differ from each other by uncorrelated random noise.” Would it be possible to verify this claim by creating statistics of the activations of neurons in the last layer?

6. Other minor issues:
    - When mentioning NTK in ln 290, I assume you are talking about neural tangent kernels. Please add proper references for that.
    - For any figures/tables in appendix, please refer then with “Appendix”, e.g. ln 245, Figure 8-c should be Appendix Figure 8-c


**Limitations:**

Yes. Limitations are well stated and discussed.


**Strengths And Weaknesses:**

* Strengths

  - Originality: the paper brought a novel insight into an old problem of deep neural networks: why overfitting does not kill the generalization capability.
  - Quality: the author(s) did comprehensive experiments and ablation studies on four datasets, pMNIST, CIFAR10, CIFAR100 and ImageNets with different variations of Wide-ResNet-28, ResNet50 and DenseNet.
  - Clarity: The paper was well written with clear explanations of background, related works, methods, experimentations and limitations.
  - Significance: The problem that the author(s) were trying to attack is very important to both the understanding of the neural network training dynamics and the design principles of the wide and deep neural network architectures in practice.

* Weaknesses:
  - Please refer to the Questions sections.

---

> ### Author Response · Authors · 2022-07-30
> **Response to reviewer aMM9**
>
> Thank you for your time reviewing the paper. We address your questions point-by-point below, and we report an additional experiment we ran following your suggestion to investigate the role of the kernel size. We hope that we can clarify your concerns. Thank you!
>
> > 1.1 Does [the work] generalize to Transformers or other networks with attention maps?
>
> We agree with the reviewer that extending the analysis to transformers would be very interesting. However, the scaling of the transformers make the analysis impossible with moderate computational resources. In a transformer, the width corresponds to the number of activations in the MLP layers. In the architectures used in practice, this number ranges from $\approx 500$ to $\approx 2000$. Just to give an example: one of the largest vision transformers, ViT-Huge (https://arxiv.org/pdf/2010.11929.pdf), has an embedding width of 1280, which is much smaller than the number of features in the hidden layer of our largest ResNet50 (8192).
> On the contrary, in convolutional networks the width can be increased by a factor 10 or more due to the small memory footprint of the 3x3 convolutional filter banks. Hence we are not able, in a transformer, to span a sufficiently large range of "widths" do fit well a scaling law.
>
> > 1.2 How much does it depend on the existence of the convolution units, which have limited span? By changing the filter size of the convolutional layers from small to very large, will the following claim hold “... neurons tend to split into groups that carry identical information”?
>
> We thank the reviewer for this suggestion and conducted an additional experiment in which we increased the kernel size from $3\times3$ to $7\times 7$ in the Wide Resnet28 and Densenet40 architectures. Preliminary results are available in the anonymous repository (https://anonymous.4open.science/r/redundant_representation2022-E4FD) in "response_rev_figures/1_error_decay_ker7.png". We find that the decay of the test error follows closely $w^-{1/2}$ also when the kernel size is 7x7. We will mention this additional result in the final version of the manuscript.
>
> > 2. Continuing training at zero training error [...] if the training error stays zero, where does the gradient come from?
>
> When the networks reaches zero training error, the number of misclassified images is zero. However, the cross-entropy loss on which we train the network remains finite, since there remains some uncertainty in the network's assignment of each image to a class. The finite loss then provides a finite gradient even when the model is interpolating.
>
> > 3. L2 norm, Gaussian priors, and other regularisations
>
> The weight decay we used is an $\ell_2$-norm penalty on the weights, and is equivalent to a MAP inference with Gaussian prior on the value of the weights (see e.g. https://www.deeplearningbook.org/contents/regularization.html paragraph 7.1.1 and end of paragraph 7.1.2).
>
> Concerning other regularizations, we tested that when weight decay is not applied, data augmentation or early stopping alone are not enough to induce a redundant representation  and a scaling with w^-0.5 (lines 219ff), highlighting the crucial role of this common regularization technique.
>
> > 4. [...] in Figure 2, there are only 2 data points per model beyond w_c=256; [...]apparently, the empirical claim of “the error of the chunks with w neurons decays as w^{-½}” is a weak claim, especially for more complex datasets CIFAR 100 and ImageNet
>
> We agree that our data is not sufficient to be sure that the scaling is W^(-1/2), so we added a series of statistical measures ($R^2$, correlation of the residuals, see comments below) to validate our main claim, which is the formation of redundant representations in wide neural networks.
> ImageNet (Fig. 8) is included in the paper exactly as an example of a case in which the scaling and all the statistical quantities used to detect the presence of redundancies are *qualitatively* different from the case of CIFAR data sets. We provide an explanation of the reason of this discrepancy at lines 231-250.

---

> > ### Author Response · Authors · 2022-07-30
> > **Continuation response to reviewer aMM9**
> >
> > > 4. [continued] Are there any other experiments or insights to support the main claim?
> >
> > Additional key evidence for this claim comes from our regression analysis, where we show that we can reconstruct the full representation using only a chunk of neurons with linear regression (Eq. 1 and Fig 4, lines 174ff). We use the $R^2$ coefficient of the linear reconstruction to measure how well a chunk of randomly selected neurons predicts the full representation. We find for various models on various data sets that for chunks of neurons that are larger than a critical size $w_c$, the explained variance $R^2$ reaches almost its theoretical maximum, namely 1. We call chunks of this size clones, since although being smaller, they contain the full information of the final representation. For example, even a subset of 128 neurons of the final representation of a Wide ResNet-28 with full width = 512 contains the full information of this representation (Fig. 4e, blue line reaches $R^2=1$ around chunk size 128). Another important statistical evidence is given by the analysis of the correlation of the residuals as discussed below.
> >
> > > 5. Statistical evidence for the claim that “distinct clones differ from each other by uncorrelated random noise.”
> >
> > This claim is also based on the regression analysis where we reconstruct the full representation using only a chunk of neurons (see our response to the previous question) The linear reconstruction is not exactly equal to the full representation, but has a reconstruction error. This error is a random vector with a number of components equal to the number of activations of the full representation, and we can analyze the correlation of the reconstruction errors of different components (lines 93-97). When we reconstruct the full representation from a chunk with a small number of neurons, say 8, we make an error because we are missing some features in the chunk that are present in the full representation. Hence, the errors of different components are correlated, because they all miss a key information. We quantify this by analyzing the covariance between these "residual" errors, for different models and data sets (cf. the green lines in Fig. 4 d-f) . As we increase the number of neurons in the chunk, we are not only able to reconstruct the full representation almost perfectly (explained variance $R^2$ approaches 1), but also with an uncorrelated residual error among different components (the green lines go to zero). This means that a clone contains essentially all the information that is in the full representation, and that they only differ from it by a small, uncorrelated noise.
> >
> > > Other minor issues
> >
> > Thank you for pointing those out, we added references to the NTK and lazy learning papers, and we changed the numbering of images in the appendix to make the distinction clearer.

---

### Author Response · Authors · 2022-07-30
**Summarising comment to the chairs**

We thank the chair and the senior area chair for their time in reviewing this paper. We put forward a mechanism which explains why adding parameters to a deep neural net that has reached zero training error will typically improve its generalisation performance. Our key idea is that networks develop redundant representations, instead of overfitting to spurious correlations. We find these redundant representations in various state-of-the-art vision models on standard benchmarks like CIFAR10/100 and analyse their properties quantitatively. We were glad to read that the reviewers found our findings "novel" (aMM9, Ef6D) and "interesting" (NAgd), the analysis "extensive" (NAgd) and the paper "well written" (aMM9).

We were able to alleviate one concern of reviewer aMM9 regarding kernel size by an additional experiment, and we clarified the relation of our results to the bias-variance trade-off with another experiment to answer an interesting question by reviewer Ef6D.

We hope that our additional results and our replies to the reviewers' comments can clarify any remaining questions.

---

### Meta-Review · Area_Chair_sc1s · 2022-08-23

**Recommendation:** Accept
**Confidence:** Certain

**Metareview:**

This paper presented a perspective into explaining why overparameterized neural networks still generalize well. By studying the statistical properties of the last layer representations, the paper showed that neural networks learn redundant representations instead of overfitting to spurious correlations. This is an important topic in understanding deep learning. The reviewers agreed that the hypothesis is clearly formulated and the empirical analysis are convincing.

**Award:**

No

---

### Decision · Program_Chairs · 2022-09-14

Accept